# Recognizing Similar Musical Instruments with YOLO Models

Christine Dewi [1] , Abbott Po Shun Chen [2,*] and Henoch Juli Christanto [3,*]

1. Department of Information Technology, Satya Wacana Christian University, 52-60 Diponegoro Rd., Salatiga 50711, Indonesia; christine.dewi@uksw.edu
2. Department of Marketing and Logistics Management, Chaoyang University of Technology, Taichung City 413310, Taiwan
3. Department of Information System, Atma Jaya Catholic University of Indonesia, Jakarta 12930, Indonesia
* Correspondence: chprosen@gm.cyut.edu.tw (A.P.S.C.); henoch.christanto@atmajaya.ac.id (H.J.C.)

**Abstract:** Researchers in the fields of machine learning and artificial intelligence have recently begun to focus their attention on object recognition. One of the biggest obstacles in image recognition through computer vision is the detection and identification of similar items. Identifying similar musical instruments can be approached as a classification problem, where the goal is to train a machine learning model to classify instruments based on their features and shape. Cellos, clarinets, erhus, guitars, saxophones, trumpets, French horns, harps, recorders, bassoons, and violins were all classified in this investigation. There are many different musical instruments that have the same size, shape, and sound. In addition, we were amazed by the simplicity with which humans can identify items that are very similar to one another, but this is a challenging task for computers. For this study, we used YOLOv7 to identify pairs of musical instruments that are most like one another. Next, we compared and evaluated the results from YOLOv7 with those from YOLOv5. Furthermore, the results of our tests allowed us to enhance the performance in terms of detecting similar musical instruments. Moreover, with an average accuracy of 86.7%, YOLOv7 outperformed previous approaches and other research results.

**Keywords:** YOLOv7; YOLOv5; similar musical instrument detection; neural network; deep learning

## 1. Introduction

Object detection is an example of computer technology which is related to computer vision. This method is used to find specific examples of semantic items that belong to a particular class, such as people [1,2], musical instruments [3,4], buildings [5], traffic signs [6], or cars [7,8], in video and digital images.

Despite the widespread application of object detection, its performance is likely to vary, depending on the possibilities. A suitable illustration of this phenomenon is provided by the condition where two object classes share the same appearance, as seen in Figure 1. Due to this, the detector is distracted from the class of object being examined. Imagine for a moment that different things that have similar external characteristics are grouped together as pairs of related objects.

Both the flute and the clarinet have several complementary characteristics. The clarinet is a woodwind device consisting of a mouthpiece with a single reed, a cylindrical tube with a flared end, and a key that conceals a hole in the tube. Both the flute and the clarinet are important members of the woodwind family of musical instruments, and they are often played concurrently. The presence or absence of reeds is one of the most important differences between the flute and the clarinet. The flute has no reed, while the clarinet has only one reed.

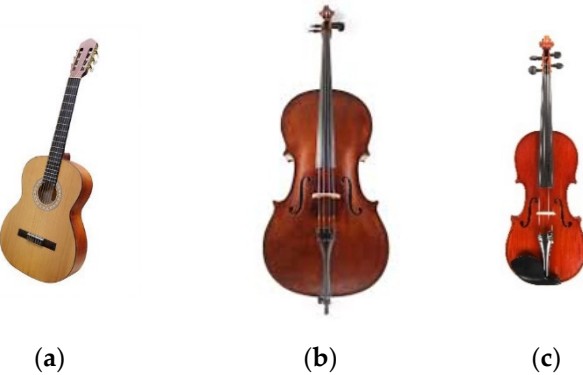

(**a**)                               (**b**)                               (**c**)

**Figure 1.** Similar musical instruments: (**a**) guitar, (**b**) cello, and (**c**) violin.

Moreover, although the cello and violin belong to the family of stringed instruments, they cannot be confused with each other in any way. The distinction in size between the cello and the violin is the main differentiator between the two instruments. When one plays the cello, it is customary to take a sitting position and hold the instrument between the knees while doing so. The violinist, on the other hand, holds the instrument so that it is supported between the shoulder and the chin. The cello can play lower notes than the violin. The cello and violin are both played with a bow, which is one thing they have in common. The right hand is used to play the cello using a bow that crosses all four strings, similar to how the violin is played. Overall, while these instruments have some similarities, they also have unique features that make them distinct from each other.

However, as can be seen in Figure 1, the designs of the guitar, violin, and cello are fundamentally very similar to one another. Computers have a far harder time differentiating among similar musical instruments than people do. The simplicity with which humans can recognize visual identification cues, such as detecting highly similar musical instrument objects, was impressive in our research study because it was one of the challenges that we investigated. In addition, even though people have no difficulty understanding the task, computers experience greater difficulty.

Recognizing similar musical instruments is important for several reasons. Firstly, in the field of musical education, by recognizing musical instruments through image detection, we can help students to learn and understand different types of instruments. This can aid in their musical education and development, especially if they do not have access to physical instruments to practice with. In the field of music production, recognizing musical instruments through image detection can help producers to create better and more engaging music. They can use this technology to identify the instruments being used in a recording or performance and make decisions on how to enhance or balance the sound. In the area of classifying instruments, image detection can be used to classify musical instruments on the basis of their physical characteristics. This can be useful in identifying and cataloguing different types of instruments and can aid in research into and preservation of musical history. In the field of performance analysis, image detection can also be used to analyze and evaluate musical performances. Through recognition of the instruments being played, it is possible to assess the quality and accuracy of a performance and provide feedback for improvement. Lastly, recognizing musical instruments through image detection can make music more accessible to those with disabilities or limitations. For example, individuals who are visually impaired can benefit from this technology, as it can provide them with a visual representation of the instruments being used in a piece of music.

Overall, recognizing musical instruments through image detection can improve musical education, production, classification, performance analysis, and accessibility. It is a valuable tool for musicians, music educators, producers, researchers, and anyone else involved in the creation and performance of music.

The most effective use of You Only Look Once (YOLO) is found in circumstances that call for faster detection. It offers a high degree of precision and a high detection rate

at the same time. YOLOv7's trainable bag of freebies represents the new state-of-the-art for real-time object detectors and is the latest version of YOLO, which was announced in 2022 [9]. In terms of both speed and accuracy, YOLOv7 is superior to any other object detector that has been developed. In this study, convolutional neural network (CNN) models and feature extractors, including YOLOv7 for object recognition, as well as other approaches to feature extraction were investigated in detail.

Our research fine-tuned the models to the People Playing Musical Instruments (PPMI) dataset [10]. The PPMI dataset contains photographs of individuals interacting with a variety of musical instruments, with 12 distinct types of instruments represented. It can be challenging to locate many object detectors in published research that have been built on deep learning and tailored specifically to the domain of detecting similar musical instruments, and thus it was difficult to locate a prior study that assessed a variety of crucial factors, such as the mAP, precision, and recall.

The following is a summary of the contributions made by this study. Firstly, we aimed to distinguish objects that appear to the human eye as very similar to each other. In the second step of our process, we applied YOLOv7 to determine which musical instruments were similar to one another. After that, we analyzed and evaluated the YOLOv7 model. Performance metrics were used to track crucial data, including the mean average precision (mAP), precision (P), and recall (R).

During this investigation, we became familiar with a wide variety of musical instruments that are similar to each other.

This article is organized as follows. Relevant prior work is presented in Section 2. Our recommended methodology is described in Section 3. In Section 4, we discuss the experimental results, and present a comprehensive analysis and interpretation of our results. At the end of the article, in Section 5, our conclusions are presented, along with recommendations for additional research.

## 2. Related Works

### 2.1. Identifying Similar Musical Instruments with CNN

Over the past few years, significant advances have been made possible by the application of deep learning to most object recognition and identification algorithms. The act of recognizing objects is simple for people, but it is quite difficult for computers to distinguish between two things that are almost identical in both their appearance and their function [11]. Two-stage detection consists of two processes that cooperate with each other to achieve the desired result. Utilizing a technique known as region-based CNN (RCNN), the detector first generates hypotheses about the possible locations of the objects in the image. This location is just a suggestion. After that, each region of interest (RoI) is classified independently, and then the classifications are combined [12].

However, two-stage detection, despite its excellent performance, has some significant drawbacks. Since there are two different processes involved, it takes a long time to train the model, and even more time to test it. To reduce the amount of time spent predicting the results, it is recommended to use a single-stage detector. YOLO [13] and the Single-Shot Detector (SSD) [14] are the most representative single-stage detectors. Compared with their two-stage equivalents, single-stage detectors are superior in terms of their overall performance, efficiency, and the number of model parameters that they require.

Ju et al. [15] described a method of object recognition that makes use of entropy loss in order to improve the ability to correctly recognize items that have a similar outward appearance. When entropy loss is applied, the detector can generate more accurate predictions regarding the bounding box class that has been observed, which ultimately leads to a higher probability of receiving a satisfactory score. In addition to this, it has the effect of reducing the deterioration of reliability. As a direct consequence, the performance in terms of detecting similar things is improved. A more effective architecture for a CNN network was created by Shijin Song and colleagues [16]. This design made it possible for small objects to be identified with greater precision, while also needing less processing

and enabling simpler deployment. They eliminated the CNN network, which significantly reduced the model's size and the amount of time the model was operational, while preserving its accuracy. To improve the effectiveness of computation, the fully convoluted layers were simultaneously replaced with fully connected layers.

Dewi et al. [17] used the YOLO approach in conjunction with the Generative Adversarial Network (GAN) in order to identify musical instruments that were comparable to one another. YOLO is a strong region-based convolutional neural network (CNN) that is extremely fast. When Deep Convolution YOLO-GAN is utilized, the capacity of the YOLO detection process will rise, even beyond what was previously possible with YOLO. In this experiment, we used the most recent version of YOLOv7 in conjunction with the PPMI dataset, which included 12 unique musical instruments in total.

*2.2. YOLOv5 and YOLOv7*

Here, we describe the timeline of the YOLO versions. The first version of YOLO, YOLOv1, was introduced in 2015. It was designed to detect objects in real time using a single neural network. YOLO9000v2 was introduced in 2016. It used a more powerful neural network architecture called Darknet-19 and introduced anchor boxes and batch normalization to improve the accuracy of object detection. Next, YOLOv3 was introduced in 2018. It introduced several new features, including a feature pyramid network, improved anchor box clustering, and multi-scale predictions. YOLOv3 achieved state-of-the-art performance on several object detection benchmarks [18]. It divided the input images into $N \times N$ grid cells [19] of the same size, and forecasted the bounding boxes and probabilities for each grid cell. YOLOv3 made use of multi-scale integration for producing predictions, and a single neural network was utilized to construct a general overview of the input. Both processes can be carried out by YOLOv3. YOLOv3 can generate a one-of-a-kind bounding box anchor for each ground truth item [20].

YOLOv4 was introduced in 2020. It introduced several advanced techniques, including CSPDarknet53, scaled-YOLOv4, PP-YOLO, YOLOv5, YOLOv6, and Mish activation. YOLOv4 achieved state-of-the-art performance on several object detection benchmarks [21]. The structure of YOLOv4 is as follows: (1) backbone: CSPDarknet53 [22], (2) neck: SPP [23] and PAN [24], and (3) head: YOLOv3 [18]. In the backbone, YOLOv4 utilizes a Mish [25] activation function. YOLOv5 was also introduced in 2020. It used a different neural network architecture called CSPNet and was designed to be faster and more accurate than previous versions of YOLO. YOLOv5 achieved state-of-the-art performance on several object detection benchmarks.

There are five unique designs for the architecture of YOLOv5, namely YOLOv5s, YOLOv5m, YOLOv5n, YOLOv5l, and YOLOv5x. The major component that distinguishes them is the number of feature extraction modules and convolution kernels that are scattered over the network at various preset points. YOLOv5 has four main components: the input, the backbone, the neck, and the output [26]. The major task of the backbone model is to identify significant segments for analysis from inside the input image.

Automatic learning bounding box anchoring, mosaic data enhancement, and cross-stage partial networking are just a few of the technologies that have been incorporated into the architecture of YOLOv5. The design makes use of the state-of-the-art algorithm optimization techniques for convolutional neural networks that have emerged in the last few years. YOLO's detection architecture is the foundation on which it was constructed. YOLOv5 uses cross-stage partial networks (CSP) and spatial pyramid pooling (pSPP) as its fundamental building blocks to extract rich, significant attributes from the input pictures. To correctly generalize a model in terms of scaling the objects, SPP is useful for detecting the same item in different sizes and scales. The feature pyramid architectures of the feature pyramid network (FPN) [27] and the path aggregation network (PANet) [28,29] were utilized in the construction of the neck network.

YOLOv6 and YOLOv7 were released in 2022 along with other methods such as DAMO YOLO and PP-YOLOE. Some researchers have created their own versions of YOLO by

modifying the existing architectures or using different backbones. YOLOv6 focused on making the system more efficient and reducing its memory footprint. It made use of a new CNN architecture called SPP-Net (spatial pyramid pooling network). This architecture is designed to handle objects with different sizes and aspect ratios, making it ideal for object detection tasks. YOLOv7 was then introduced. One of the key improvements in YOLOv7 is the use of a new CNN architecture called ResNeXt.

The YOLOv7 method has caused a sensation in the fields of computer vision and machine learning. When compared with other object detection models and older YOLO versions, the newest YOLO algorithm is much faster and more accurate. Moreover, YOLOv7 is a variant of the YOLO (You Only Look Once) object detection algorithm, which was first introduced by Joseph Redmon et al. in 2016 [18]. YOLOv7 is based on a deep neural network and is capable of detecting and localizing objects within an image in real time. Compared with earlier versions of YOLO, YOLOv7 has several improvements, including the use of skip connections and the introduction of residual blocks in the network architecture. These changes allow YOLOv7 to detect objects with greater accuracy and speed. In order for the YOLOv7 method to function properly, the input image must first be segmented into a grid of cells. Next, the algorithm must predict the bounding boxes and class probabilities of the objects contained within each cell. Each bounding box's prediction includes a confidence score, indicating the probability that the predicted box contains an object. The algorithm also predicts the class probabilities for each object, which are used to label the objects within the bounding boxes. Furthermore, YOLOv7 is a popular algorithm for object detection tasks due to its high speed and accuracy. It has many applications, including in autonomous vehicles, surveillance systems, and robotics. YOLOv8 is the latest version of YOLO and was released in 2023. However, Ultralytics YOLOv8 provides the most advanced capabilities and has outperformed previous versions. Ultralytics YOLOv8 provides a unified framework for training models for the tasks of object detection, instance segmentation, and image classification. This means that users can use a single model for all three tasks, simplifying the training process.

In addition to being able to be trained significantly more quickly on tiny datasets without any pre-learned weights, it also requires technology that is several times cheaper than other neural networks. YOLOv7's architecture is a combination of YOLOv4, scaled YOLOv4, and YOLO-Rs, among others. To produce a new and enhanced version of YOLOv7, more tests were carried out using these models as a foundation. The extended efficient layer aggregation network (E-ELAN) serves as the computing node of YOLOv7's backbone [30,31]. It was based on earlier studies on how effective networks are. It was developed by considering the following elements that affect speed and accuracy, such as the memory access cost, the ratio of the input to the output channel, element-wise operation, activations, and the gradient path. Expansion, shuffling, and merging the cardinality are the three techniques that the proposed E-ELAN uses in order to keep the initial gradient path intact, while continuously enhancing the network's capacity for learning. In order to make YOLOv7 better, a compound model scaling technique was utilized. Within this framework, the width and depth of models based on concatenation can be scaled in a consistent manner.

Bag of freebies (BoF) techniques improve a model's output without adding to its training budget. The following are some of the new BoF techniques included in YOLOv7. First, planned re-parameterized convolution, which improves a model by applying re-parameterization, is a common practice following training [32]. The model takes longer to train but yields better inference results [33]. Model-level and module-level ensemble re-parametrization were used to construct the models. Second, with the "coarse for auxiliary and fine for lead loss" technique, the YOLO architecture comprises of a backbone, a neck, and a head. The outputs that were anticipated are stored in the head. YOLOv7 is not restricted to utilizing a single head at a time [34]. It can accomplish everything it wants since it possesses several heads. Moreover, the lead head-guided label assigner encapsulates the concepts of the lead head, the auxiliary head, and the soft label assigner. The YOLOv7

network's lead head is the part responsible for making the final predictions. These results serve as the basis for the generation of soft labels. The loss is calculated for both the lead head and the auxiliary head on the basis of the same soft labels being created, which is crucial to take into consideration. Ultimately, both heads will be trained using the soft labels. Further, the coarse-to-fine labels include a fine label to train the lead head and a set of coarse labels to train the auxiliary head.

YOLO is a popular object detection algorithm that is widely used in computer vision applications. YOLO has several versions, with YOLOv5 and YOLOv7 being the latest ones. YOLOv5 is a later version of the algorithm and has several improvements over its predecessor, YOLOv4. Some of the key features of YOLOv5 include its improved speed and improved accuracy, and that it is a smaller model. In terms of speed, YOLOv5 is faster than YOLOv4, allowing real-time object detection in video streams. In terms of accuracy, YOLOv5 has better accuracy compared with previous versions, with a better ability to detect small objects and objects at a distance. As a lighter model, YOLOv5 has a smaller model compared with previous versions, making it easier to deploy on embedded devices and systems with limited resources. On the other hand, YOLOv7 is a more recent development. The main difference between YOLOv5 and YOLOv7 is that YOLOv7 uses a different architecture from YOLOv5. YOLOv7 uses an anchor-free architecture, which eliminates the need for anchor boxes, which can improve the accuracy of object detection. YOLOv7 also uses a different backbone network, which can improve the speed and efficiency of the algorithm. Overall, both YOLOv5 and YOLOv7 are powerful object detection algorithms that can be used for a wide range of applications. The choice between the two depends on the specific requirements of the application, such as the need for speed and/or accuracy, or the model's size.

In our experiment, we implemented YOLOv5n, YOLOv5s, YOLOv5m, YOLOv7, and YOLOv7x, as described in Table 1.

**Table 1.** An overview of the YOLOv5 and YOLOv7 models used with the COCO dataset.

| Model Name | Accuracy (mAP 0.5) | Params (Million) | GPU Time (ms) | CPU Time (ms) |
|---|---|---|---|---|
| YOLOv5n | 45.7 | 1.9 | 6.3 | 45 |
| YOLOv5s | 56.8 | 7.2 | 6.4 | 98 |
| YOLOv5m | 64.1 | 21.2 | 8.2 | 224 |
| YOLOv7 | 51.4 | 36.9 | - | - |
| YOLOv7x | 53.1 | 71.3 | - | - |

## 3. Methodology

### 3.1. Dataset

Pictures of individuals posing with a wide variety of musical instruments can be found within the People Playing Musical Instruments (PPMI) dataset. Bassoons, cellos, clarinets, French horns, erhus, flutes, guitars, harps, saxophones, trumpets, recorders, and violins are among the instruments in the dataset. Yao gathered images of musical instruments and published them in [10]. In September 2010, Aditya Khosla published a collection of photos of various musical instruments that he had taken. These photos include cellos, clarinets, harps, recorders, and trumpets. Initially, the dataset contained 100 examples from each category to be used for training, as well as 100 images to use for testing. The first table provides an overview of the entire dataset. During this study, we used the PPMI dataset to train and validate our model. Figure 2 shows the PPMI dataset we used in these experiments.

The labels of the PPMI dataset are depicted in Figure 3. The PPMI dataset includes 12 classes, and the vast majority of those classes have more than 300 photos. The values for x and y can vary anywhere from 0.0 to 1.0, while the width can be any number between 0.0 and 0.1, and the height can be any value between 0.0 and 0.1. We used 70% of it for training

purposes and 30% for testing. Table 2 presents the distribution of the PPMI dataset in its entirety.

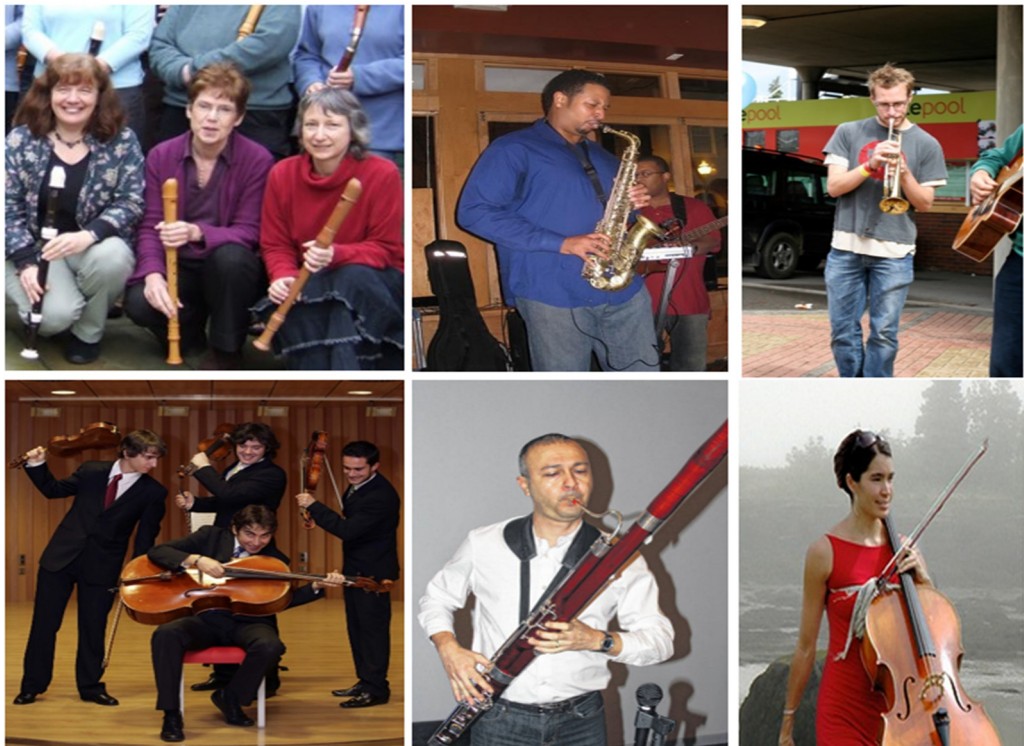

**Figure 2.** Sample images in the PPMI dataset.

**Table 2.** Distribution of the PPMI dataset.

| Class Name | Testing | Training | Total Images |
|---|---|---|---|
| Bassoon | 109 | 253 | 362 |
| Cello | 97 | 225 | 322 |
| Clarinet | 95 | 221 | 316 |
| Erhu | 101 | 236 | 337 |
| Flute | 95 | 221 | 315 |
| French horn | 98 | 229 | 327 |
| Guitar | 98 | 228 | 326 |
| Harp | 100 | 232 | 332 |
| Recorder | 93 | 216 | 309 |
| Saxophone | 98 | 228 | 326 |
| Trumpet | 99 | 231 | 330 |
| Violin | 102 | 238 | 340 |
| Total Images | 1183 | 2759 | 3942 |

### 3.2. YOLOv7

In these sections, we explain our proposed YOLOv7 architecture, as shown in Figure 4. The PPMI dataset was used as the input in our systems, and then we trained the model with YOLOv7. YOLOv7's E-ELAN architecture uses "expand, shuffle, and merge cardinality" to acquire the ability to continuously increase the network's learning capability without losing the original gradient path, allowing the model to learn better. Compound model scaling, which is based on concatenation, is new in YOLOv7. The compound scaling approach preserves the model's characteristics from the time of its inception, allowing for the most efficient architecture to be kept. Re-parameterization of the model is scheduled to take place in the architecture of YOLOv7. The RepConv of a layer that has concatenation or residual connections should not have an identity connection. Due to this, the RepConv of

that layer is replaced by RepConvN, which does not have any identity connections. The next step is to generate the final model by averaging their relative weights. The model's weights from different times should be averaged out. In our research, we examined both the training procedure and the testing procedure in depth for object recognition.

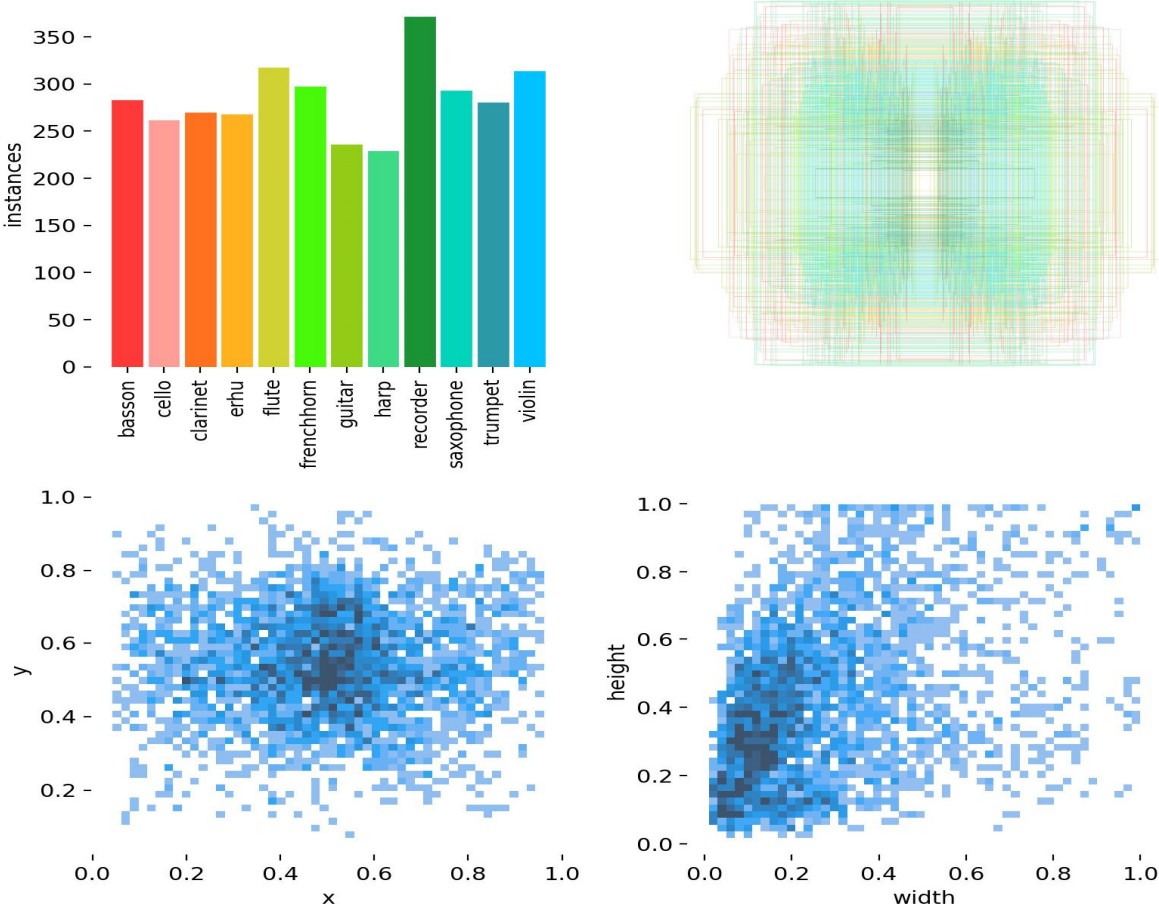

**Figure 3.** Instances in the PPMI dataset.

RepConvN, which has no identification ties, can be used in its place under certain conditions. RepConv is a convolutional layer that includes $3 \times 3$ convolutions, $1 \times 1$ convolution, and identity connections. The authors used RepConv without an identity connection (RepConvN) to create the architecture of the planned re-parameterized convolution after examining the combination and corresponding performance of RepConv and other architectures. In this study, when a convolutional layer including residuals or concatenations is swapped for re-parameterized convolution, there should be no identity connections [35,36].

The following is a list of the key differences between the various basic versions of YOLOv7. YOLOv7 is the foundational model, and it was designed to be as efficient as possible for general GPU computing. YOLOv7x was developed through the implementation of the suggested compound scaling method. YOLOv7-tiny is a fundamental model that has been tailored specifically for edge GPU. YOLOv7-W6 is a basic model optimized for cloud GPU computing. In our work, we only focused on YOLOv7 and YOLOv7x, and tuned them with the PPMI datasets. During the training of YOLOv7, we set the following parameters: image size = 640 × 640, conf-thres = 0.25, iou_thres = 0.45, learning_rate = 0.0001, number of classes = 12, depth_multiple = 1.0, momentum = 0.999, optimizer = Adam with lr0, epoch = 100, and width_multiple = 1.0. As a comparison, we also trained and tested our model with YOLOv5. Throughout the training of YOLOv5, we specified the following parameters: picture size = 640 × 640, conf-thres = 0.25, iou_thres = 0.45,

learning rate = 0.0001, max_det = 1000, number of classes = 12, depth_multiple = 0.33, momentum = (0.3, 0.6, 0.98), batch size = 16, epoch = 100, and width_multiple = 0.50.

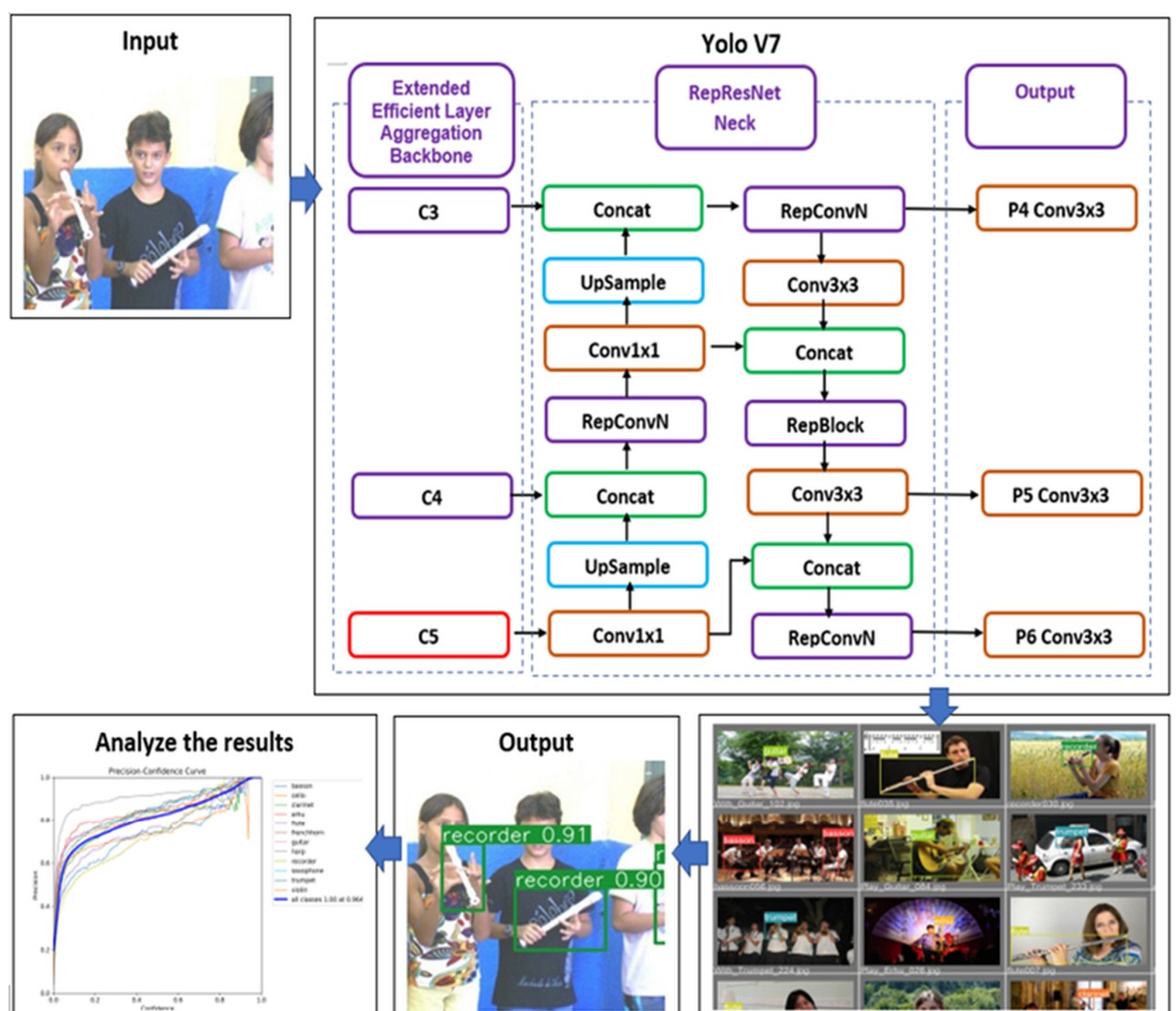

**Figure 4.** Architecture of YOLOv7.

### 3.3. Training Results

During every stage of the training process, this study made use of several data augmentation techniques, including padding, cropping, and horizontal flipping, among others. These methods are often used in the creation of large neural networks because of their benefits. Diagrams of the training process and the validation process for Batch 0 are shown in Figures 5 and 6, respectively. The YOLOv7 network augments its training material with random splicing of four images using mosaic data augmentation. This greatly increases the detection dataset, strengthens the network, and releases more video processing power on the GPU. In addition, a Nvidia RTX3060Ti GPU accelerator with 11 GB of RAM, an i7 central processing unit (CPU), and 16 GBDDR2 memory comprised the environment of the training model. YOLOv7's primary goal was real-time detection, and it was trained using only a single graphics processing unit (GPU).

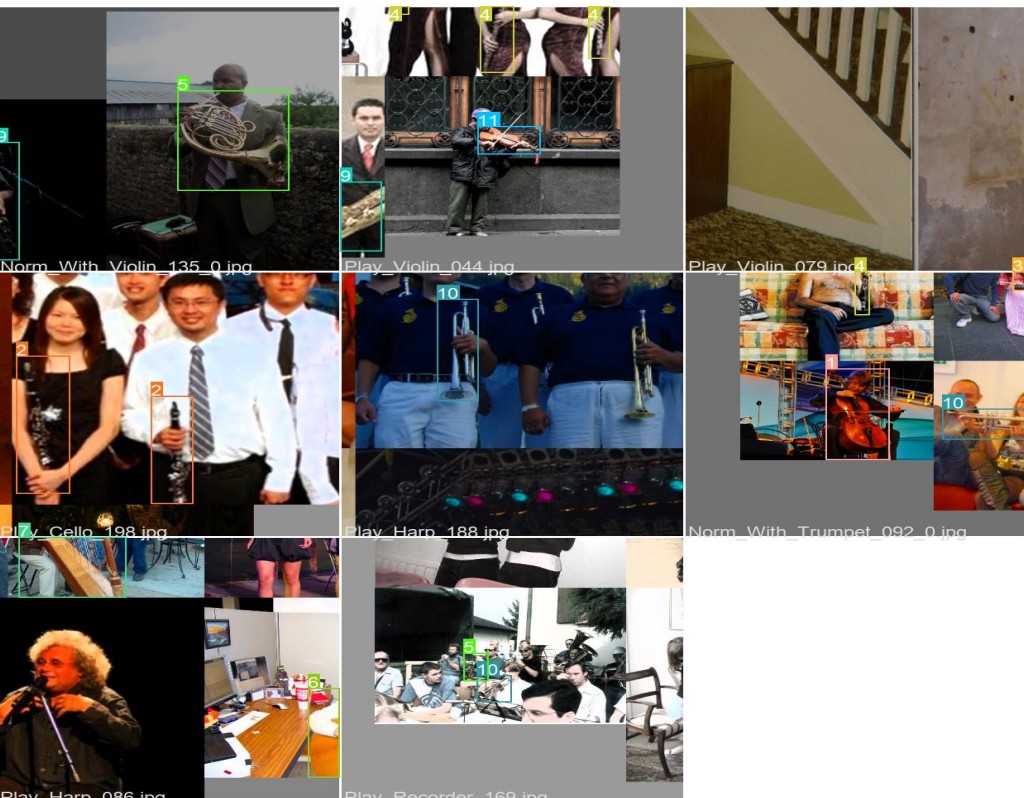

**Figure 5.** Training process: Example of Batch 0.

The BBox marking tool [37] was implemented to create a bounding box for all musical instruments. YOLO labeling is the standard output format for most annotation programs, and it creates a single text file that contains all the annotations for all images. Each text file has one bounding box, abbreviated as "BBox", and the annotation for each of the objects displayed in the image. The image-appropriate scaling of the annotations yields a value ranging from 0 to 1 for all of the labels [38]. Equations (1)–(6) were the basis for the procedure of adjustment for calculating the YOLO format:

$$dw = 1/W \tag{1}$$

$$x = \frac{(x_1 + x_2)}{2} \times dw \tag{2}$$

$$dh = 1/H \tag{3}$$

$$y = \frac{(y_1 + y_2)}{2} \times dh \tag{4}$$

$$w = (x_2 - x_1) \times dw \tag{5}$$

$$h = (y_2 - y_1) \times dh \tag{6}$$

where $H$ is used to denote the height of the image, $dh$ refers to the absolute height of the image, $W$ is used to denote the width of the image, and $dw$ represents the absolute width of the image. The results of training for all classes are shown in Tables 3 and 4, which includes the mAP, precision, and recall for each class. According to Table 3, YOLOv7x achieved a mAP of 88.2%, followed by YOLOv7, with a mAP of 86.6%. In addition, the cello, erhu, guitar, harp, and saxophone classes all earned a maximum mAP score of over 90% when

YOLOv7x was used. The precision and recall curve for YOLOv7x is displayed in Figure 7a, and the training and validation curve can be seen in Figure 7b. In addition, the harp, erhu, and saxophone classes had the highest mAP (91.1%, 97.6%, and 91.6%, respectively) with YOLOv7. As a comparison, we also trained YOLOv5, and the results are depicted in Table 4. YOLOv5m achieved the highest mAP of 82.5%, followed by YOLOv5s with 81.3% and YOLOv5n with 75%.

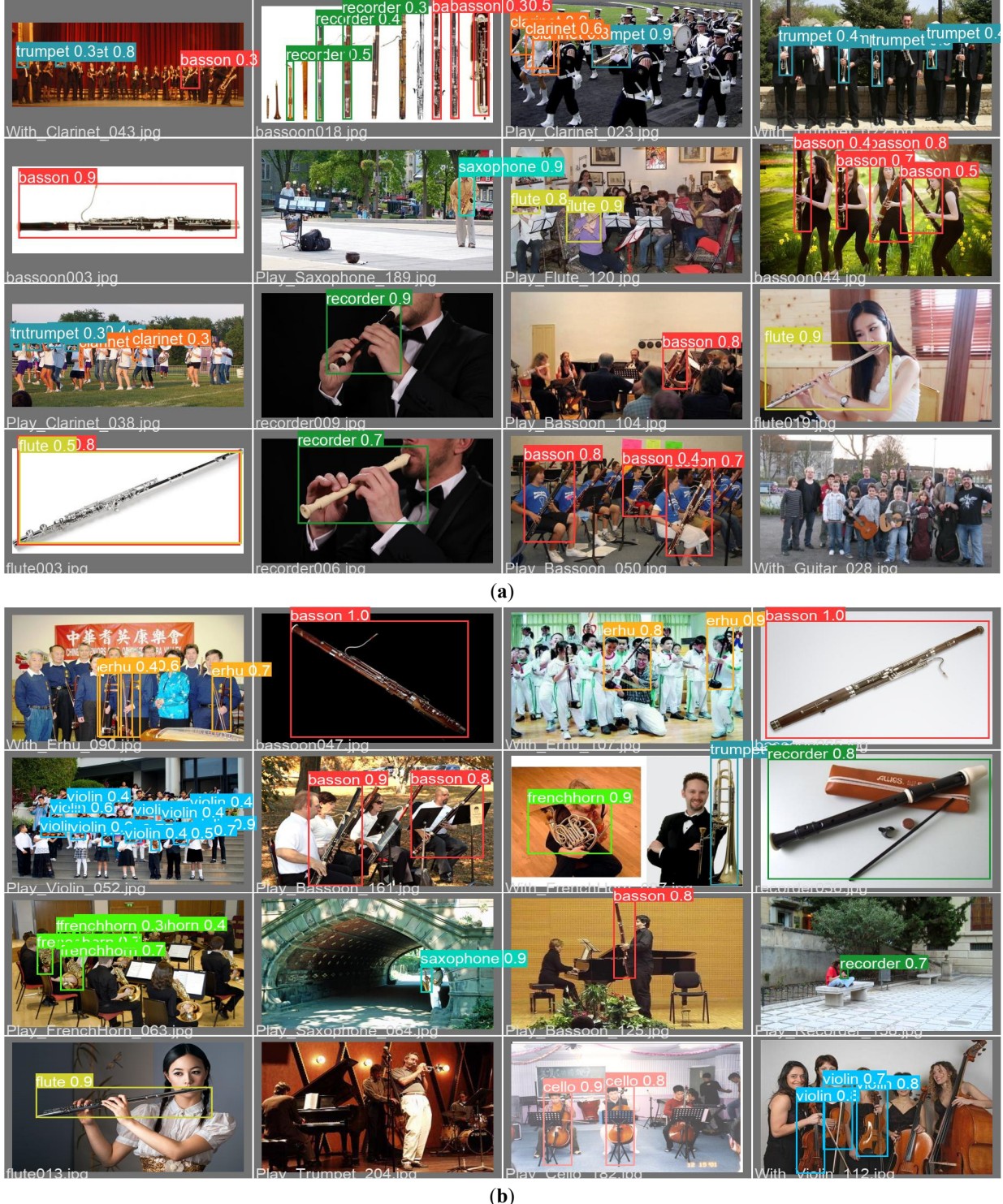

**Figure 6.** Validation process for Batch 0. (**a**) Labels and (**b**) predictions.

**Table 3.** Evaluation of the performance of training on the PPMI dataset with YOLOv7.

| Class | Images | Labels | YOLOv7 | | | YOLOv7x | | |
|---|---|---|---|---|---|---|---|---|
| | | | P | R | mAP@.5 | P | R | mAP@.5 |
| All | 1182 | 1543 | 0.795 | 0.843 | 0.866 | 0.804 | 0.84 | **0.882** |
| Bassoon | 1182 | 140 | 0.838 | 0.812 | 0.856 | 0.858 | 0.82 | 0.868 |
| Cello | 1182 | 119 | 0.735 | 0.857 | 0.869 | 0.821 | 0.832 | 0.913 |
| Clarinet | 1182 | 112 | 0.775 | 0.862 | 0.853 | 0.72 | 0.786 | 0.822 |
| Erhu | 1182 | 123 | 0.84 | 0.935 | 0.911 | 0.806 | 0.959 | 0.92 |
| Flute | 1182 | 131 | 0.762 | 0.806 | 0.826 | 0.75 | 0.779 | 0.816 |
| French horn | 1182 | 119 | 0.687 | 0.866 | 0.862 | 0.767 | 0.84 | 0.897 |
| Guitar | 1182 | 112 | 0.856 | 0.821 | 0.885 | 0.887 | 0.786 | 0.904 |
| Harp | 1182 | 110 | 0.938 | 0.963 | 0.976 | 0.955 | 0.958 | 0.988 |
| Recorder | 1182 | 173 | 0.75 | 0.798 | 0.808 | 0.721 | 0.792 | 0.816 |
| Saxophone | 1182 | 137 | 0.839 | 0.861 | 0.916 | 0.826 | 0.901 | 0.911 |
| Trumpet | 1182 | 130 | 0.743 | 0.715 | 0.804 | 0.772 | 0.792 | 0.858 |
| Violin | 1182 | 137 | 0.783 | 0.817 | 0.827 | 0.765 | 0.832 | 0.868 |

**Table 4.** Evaluation of the performance of training on the PPMI dataset with YOLOv5.

| Class | Images | Labels | YOLOv5m | | | YOLOv5n | | | YOLOv5s | | |
|---|---|---|---|---|---|---|---|---|---|---|---|
| | | | P | R | mAP@.5 | P | R | mAP@.5 | P | R | mAP@.5 |
| All | 1314 | 1748 | 0.798 | 0.825 | 0.825 | 0.725 | 0.735 | 0.75 | 0.761 | 0.802 | 0.813 |
| Bassoon | 1314 | 149 | 0.837 | 0.828 | 0.869 | 0.828 | 0.745 | 0.8 | 0.818 | 0.785 | 0.811 |
| Cello | 1314 | 124 | 0.811 | 0.887 | 0.896 | 0.772 | 0.831 | 0.816 | 0.737 | 0.871 | 0.841 |
| Clarinet | 1314 | 136 | 0.836 | 0.757 | 0.793 | 0.693 | 0.625 | 0.649 | 0.781 | 0.733 | 0.809 |
| Erhu | 1314 | 135 | 0.83 | 0.889 | 0.913 | 0.743 | 0.785 | 0.804 | 0.782 | 0.85 | 0.864 |
| Flute | 1314 | 163 | 0.727 | 0.785 | 0.799 | 0.746 | 0.62 | 0.69 | 0.708 | 0.767 | 0.733 |
| French horn | 1314 | 140 | 0.844 | 0.812 | 0.902 | 0.713 | 0.829 | 0.84 | 0.8 | 0.836 | 0.877 |
| Guitar | 1314 | 123 | 0.806 | 0.756 | 0.819 | 0.784 | 0.715 | 0.799 | 0.805 | 0.748 | 0.784 |
| Harp | 1314 | 114 | 0.913 | 0.974 | 0.982 | 0.883 | 0.93 | 0.949 | 0.907 | 0.956 | 0.969 |
| Recorder | 1314 | 209 | 0.702 | 0.745 | 0.782 | 0.57 | 0.565 | 0.572 | 0.649 | 0.703 | 0.721 |
| Saxophone | 1314 | 144 | 0.815 | 0.875 | 0.902 | 0.712 | 0.84 | 0.83 | 0.751 | 0.879 | 0.876 |
| Trumpet | 1314 | 140 | 0.729 | 0.721 | 0.733 | 0.606 | 0.586 | 0.561 | 0.718 | 0.692 | 0.698 |
| Violin | 1314 | 171 | 0.722 | 0.865 | 0.828 | 0.645 | 0.754 | 0.685 | 0.683 | 0.801 | 0.771 |

The loss function of YOLO was based on Equation (7) [13]:

$$
\lambda_{coord} \sum_{i=0}^{s^2} \sum_{j=0}^{B} \mathbb{1}_{ij}^{obj} \left[ (x_i - \hat{x}_i)^2 + (y - \hat{y}_i)^2 \right] + \lambda_{coord} \sum_{i=0}^{s^2} \sum_{j=0}^{B} \mathbb{1}_{ij}^{obj} [(\sqrt{w_i} -
$$
$$
\sqrt{\hat{w}_i})^2 + \left( \sqrt{h_i} - \sqrt{\hat{h}_i} \right)^2 \right] + \sum_{i=0}^{s^2} \sum_{j=0}^{B} \mathbb{1}_{ij}^{obj} (C_i - \hat{C}_i)^2 + \tag{7}
$$
$$
\lambda_{noobj} \sum_{i=0}^{s^2} \sum_{j=0}^{B} \mathbb{1}_{ij}^{noobj} (C_i - \hat{C}_i)^2 + \sum_{i=0}^{s^2} \mathbb{1}_{i}^{obj} \sum_{c \epsilon classes} (p_i \copyright \hat{p}_i(c))^2
$$

where $\mathbb{1}_{ij}^{obj}$ indicates that the object appears in cell $i$, and $\mathbb{1}_{ij}^{obj}$ indicates that the $j$th bounding box predictor in cell $i$ is responsible for the prediction. Next, $\left( \hat{x}, \hat{y}, \hat{w}, \hat{h}, \hat{c}, \hat{p} \right)$ were used to express the anticipated bounding box's center coordinates, width, height, confidence, and category probability. Moreover, our experiment defined the $\lambda_{coord}$ as 0.5, indicating that the errors in width and height were less useful in the computation. In order to lessen the effect of multiple grids, a loss value that is empty of objects, $\lambda_{noobj} = 0.5$, was utilized.

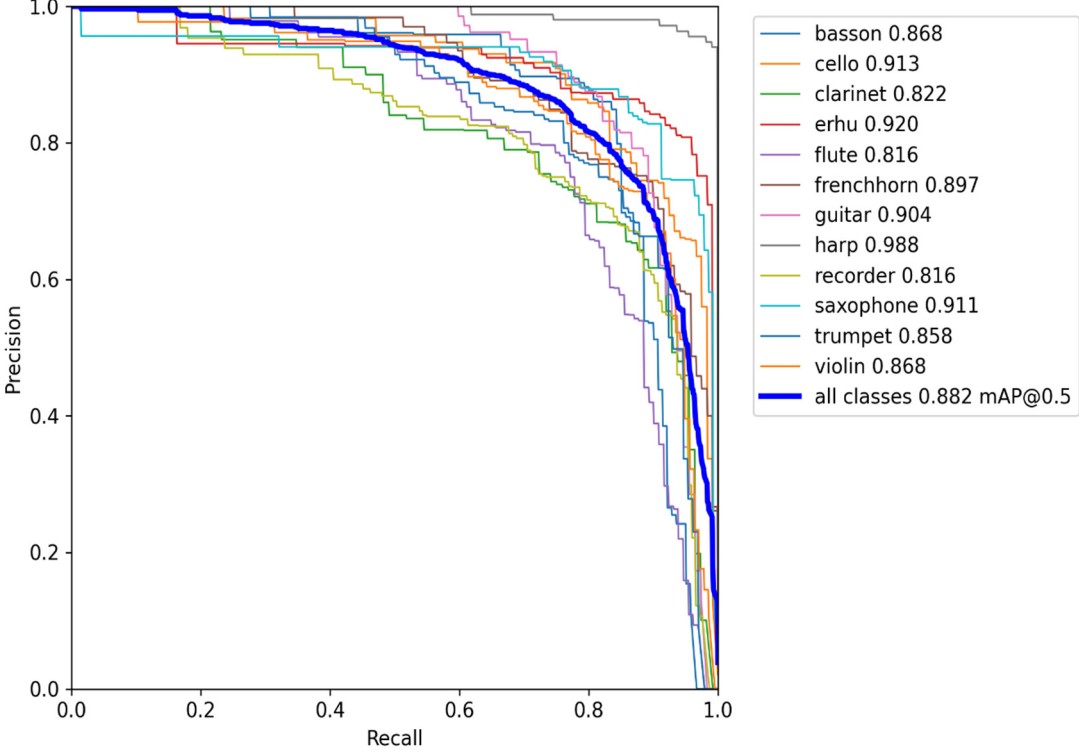

(**a**)

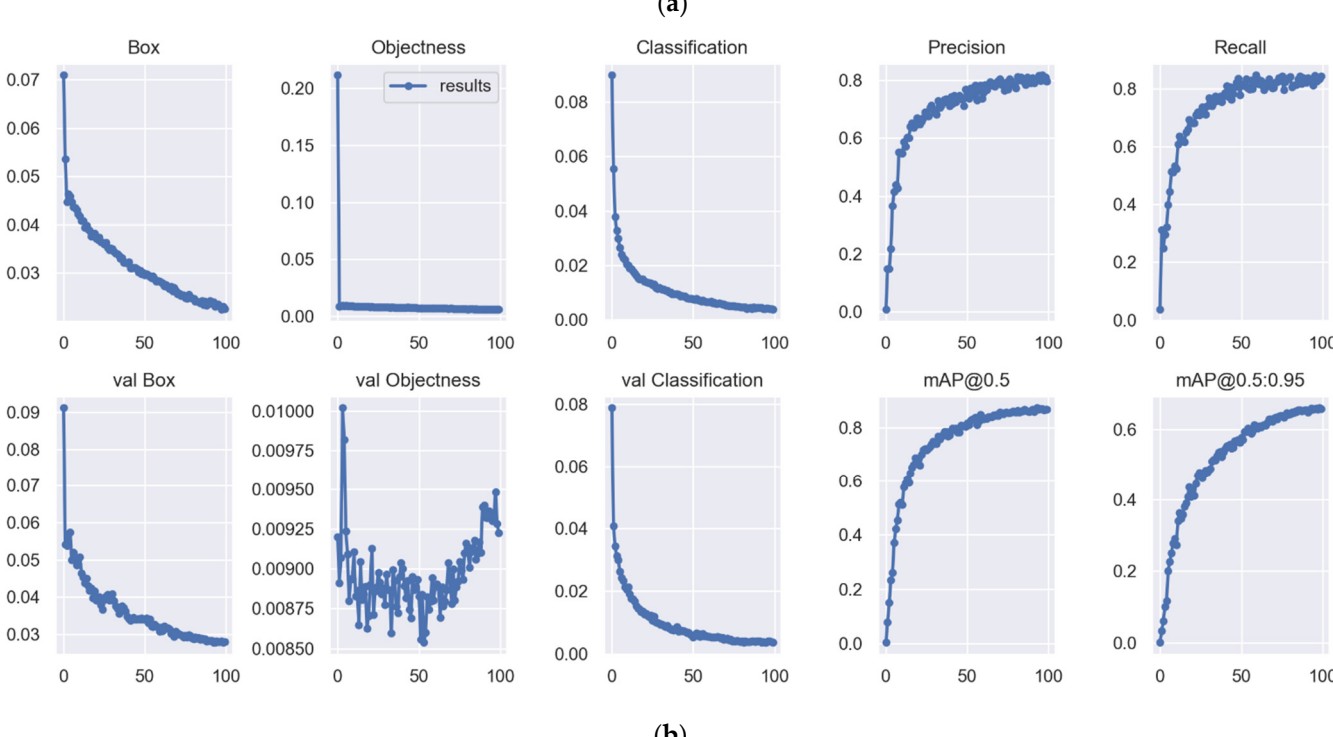

(**b**)

**Figure 7.** The YOLOv7x curves. (**a**) Precision and recall curves for YOLOv7x, and (**b**) training and validation curves for YOLOv7x.

Equation (8) describes the average mean average precision (*mAP*) as the integral over the precision *p*(*o*):

$$mAP = \int_0^1 p(0)do \tag{8}$$

where *p*(*o*) is the precision of object detection. *IoU* calculates the overlap ratio between the boundary box of the prediction (*pred*) and the ground truth (*gt*) and is shown in Equation (9). Precision and recall were calculated by Equations (10) and (11) [39].

$$IoU = \frac{Area_{pred} \cap Area_{gt}}{Area_{pred} \cup Area_{gt}} \tag{9}$$

$$Precision = \frac{TP}{TP + FP} = TP/N \tag{10}$$

$$Recall = \frac{TP}{TP + FN} \tag{11}$$

where *N* is the number of objects found, *TP* is the true positives, *FP* is the false positives, and *FN* is the false negatives (including true positives and false positives). Another evaluation index, *F1* [40], is shown in Equation (12). Figure 8 describes the confusion matrix of YOLOv7x.

$$F1 = \frac{2 \times Precision \times Recall}{Precision + Recall} \tag{12}$$

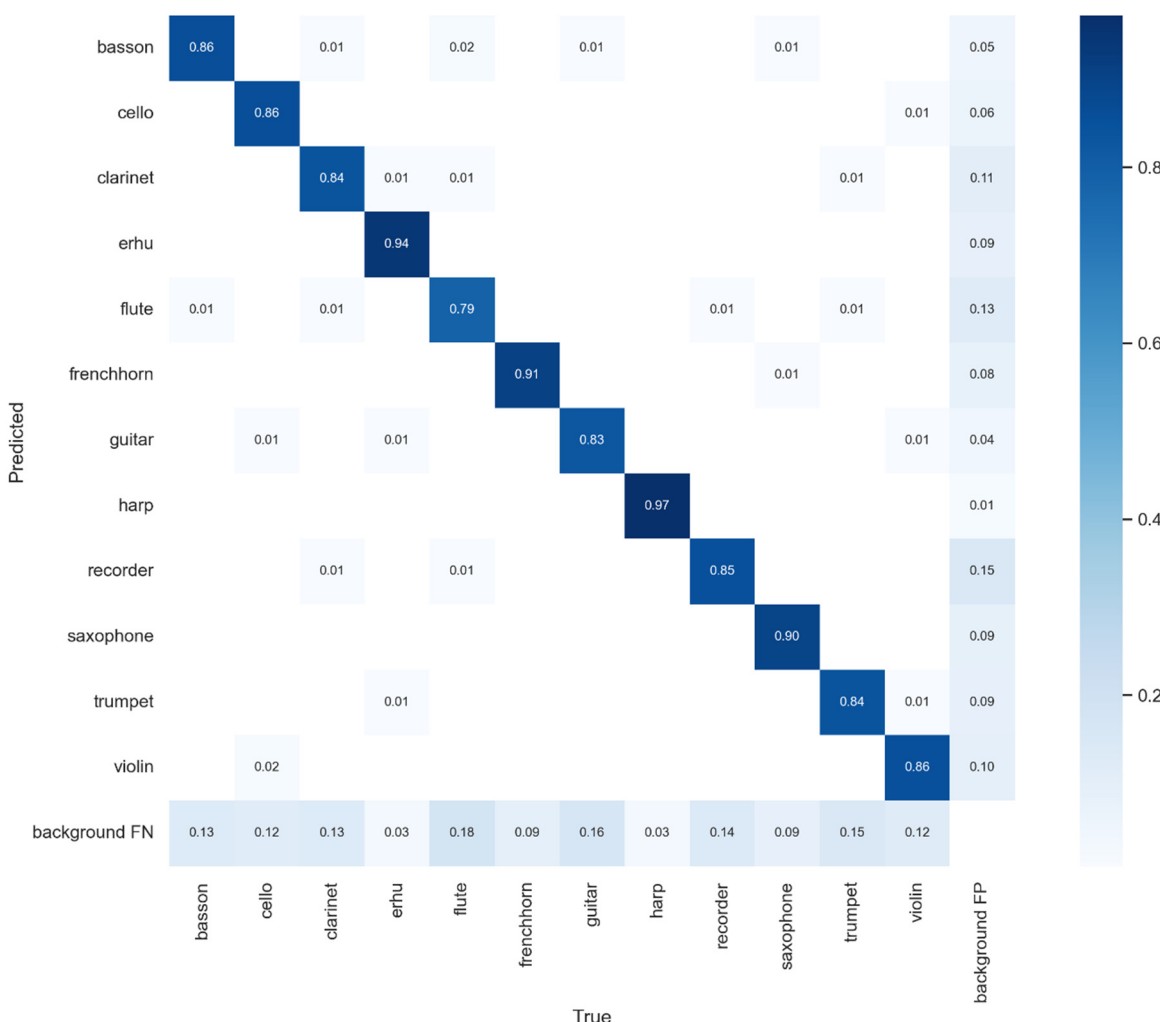

**Figure 8.** Confusion matrix for YOLOv7x.

## 4. Results and Discussion

The test results of YOLOv7's performance are detailed in Table 5, whereas the results for YOLOv5 are shown in Table 6. Our results showed that YOLOv7 achieved the highest

average mAP (86.7%), followed by YOLOv7x (86.1%), YOLOv5m (80.5%), YOLOv5s (72.6%), and YOLOv5n (64.) The harp class had the optimum mAP of 97.3% using YOLOv7 during the testing process.

**Table 5.** The performance of YOLOv7.

| Class | Images | Labels | YOLOv7 | | | YOLOv7x | | |
|---|---|---|---|---|---|---|---|---|
| | | | P | R | mAP@.5 | P | R | mAP@.5 |
| All | 1182 | 1543 | 0.808 | 0.833 | **0.867** | 0.771 | 0.826 | 0.861 |
| Bassoon | 1182 | 140 | 0.826 | 0.816 | 0.853 | 0.876 | 0.757 | 0.849 |
| Cello | 1182 | 119 | 0.751 | 0.849 | 0.862 | 0.778 | 0.823 | 0.871 |
| Clarinet | 1182 | 112 | 0.749 | 0.857 | 0.821 | 0.678 | 0.772 | 0.774 |
| Erhu | 1182 | 123 | 0.832 | 0.927 | 0.91 | 0.784 | 0.946 | 0.905 |
| Flute | 1182 | 131 | 0.799 | 0.771 | 0.831 | 0.663 | 0.802 | 0.814 |
| French horn | 1182 | 119 | 0.757 | 0.849 | 0.871 | 0.673 | 0.84 | 0.873 |
| Guitar | 1182 | 112 | 0.842 | 0.806 | 0.877 | 0.836 | 0.776 | 0.85 |
| Harp | 1182 | 110 | 0.951 | 0.955 | 0.973 | 0.95 | 0.863 | 0.968 |
| Recorder | 1182 | 173 | 0.807 | 0.751 | 0.835 | 0.716 | 0.815 | 0.801 |
| Saxophone | 1182 | 137 | 0.853 | 0.861 | 0.912 | 0.786 | 0.885 | 0.9 |
| Trumpet | 1182 | 130 | 0.738 | 0.736 | 0.811 | 0.767 | 0.784 | 0.853 |
| Violin | 1182 | 137 | 0.789 | 0.818 | 0.844 | 0.748 | 0.854 | 0.871 |

**Table 6.** The performance of YOLOv5.

| Class | Images | Labels | YOLOv5m | | | YOLOv5n | | | YOLOv5s | | |
|---|---|---|---|---|---|---|---|---|---|---|---|
| | | | P | R | mAP@.5 | P | R | mAP@.5 | P | R | mAP@.5 |
| All | 1314 | 1748 | 0.739 | 0.793 | **0.805** | 0.61 | 0.689 | 0.64 | 0.688 | 0.752 | 0.726 |
| Bassoon | 1314 | 149 | 0.762 | 0.745 | 0.791 | 0.69 | 0.626 | 0.643 | 0.706 | 0.664 | 0.719 |
| Cello | 1314 | 124 | 0.743 | 0.782 | 0.772 | 0.633 | 0.653 | 0.616 | 0.656 | 0.758 | 0.691 |
| Clarinet | 1314 | 136 | 0.751 | 0.775 | 0.784 | 0.583 | 0.684 | 0.618 | 0.7 | 0.755 | 0.727 |
| Erhu | 1314 | 135 | 0.817 | 0.829 | 0.866 | 0.676 | 0.622 | 0.681 | 0.71 | 0.817 | 0.776 |
| Flute | 1314 | 163 | 0.637 | 0.804 | 0.767 | 0.563 | 0.644 | 0.568 | 0.707 | 0.748 | 0.696 |
| French horn | 1314 | 140 | 0.826 | 0.849 | 0.895 | 0.597 | 0.847 | 0.747 | 0.723 | 0.82 | 0.79 |
| Guitar | 1314 | 123 | 0.76 | 0.675 | 0.747 | 0.695 | 0.629 | 0.659 | 0.715 | 0.675 | 0.688 |
| Harp | 1314 | 114 | 0.861 | 0.912 | 0.914 | 0.752 | 0.825 | 0.824 | 0.851 | 0.746 | 0.839 |
| Recorder | 1314 | 209 | 0.658 | 0.732 | 0.739 | 0.483 | 0.67 | 0.535 | 0.59 | 0.718 | 0.659 |
| Saxophone | 1314 | 144 | 0.743 | 0.854 | 0.873 | 0.617 | 0.778 | 0.702 | 0.656 | 0.847 | 0.797 |
| Trumpet | 1314 | 140 | 0.628 | 0.714 | 0.709 | 0.505 | 0.571 | 0.478 | 0.59 | 0.664 | 0.595 |
| Violin | 1314 | 171 | 0.681 | 0.849 | 0.798 | 0.529 | 0.719 | 0.615 | 0.65 | 0.813 | 0.734 |

Figure 9 shows the results of recognition for all models in the experiment. According to these results, all models could detect all objects in the images very well, except that YOLOv7x and YOLOv5m failed to detect all the cellos in Figure 9b,c. In addition, in Figure 9a, YOLOv7 could detect the erhu class with mAP values of 75%, 71%, and 76%. The flute class had a mAP of 92% and 76%, and of 57% and 81%.

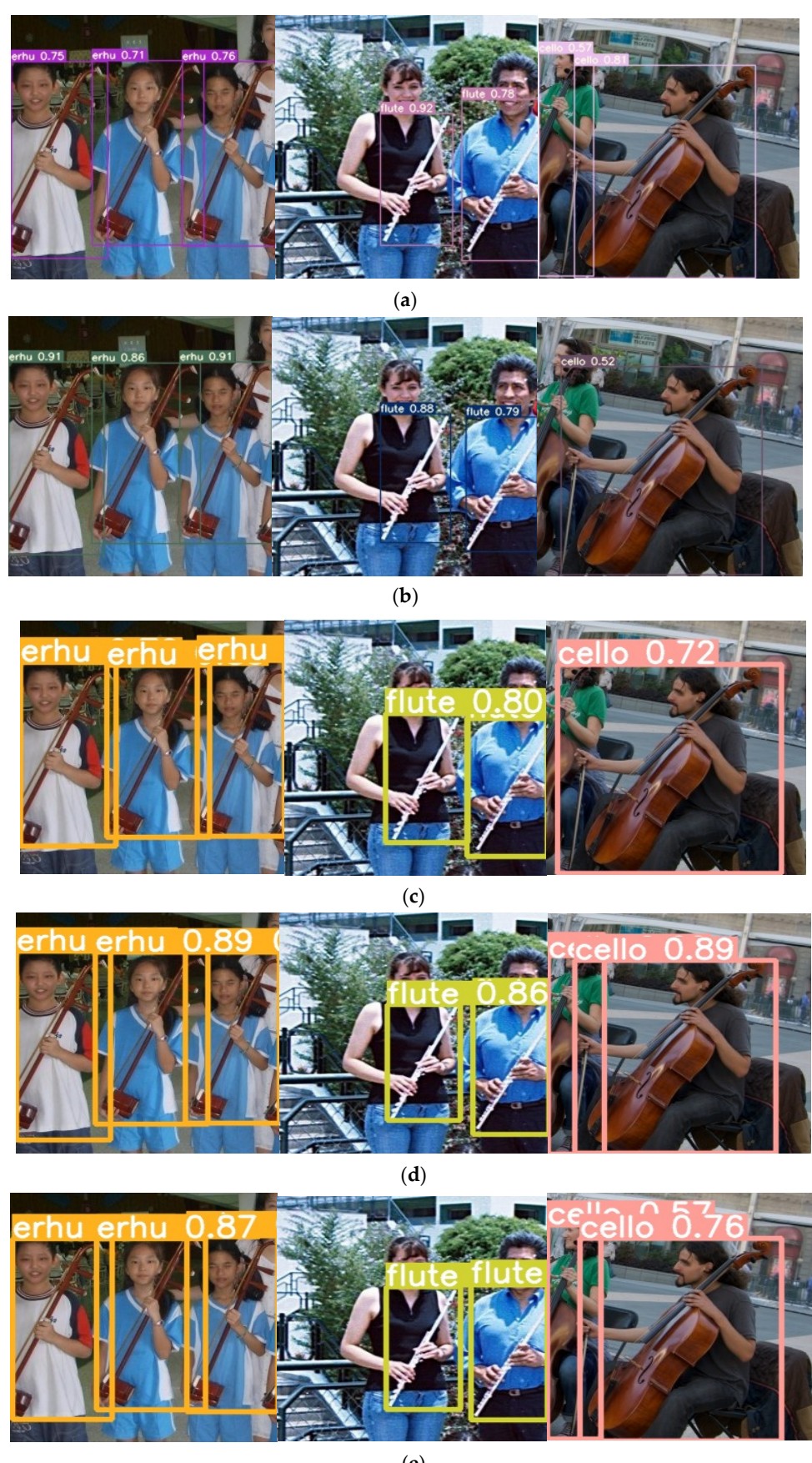

**Figure 9.** Results of recognition. (**a**) YOLOv7, (**b**) YOLOv7x, (**c**) YOLOv5n, (**d**) YOLOv5s, and (**e**) YOLOv5m.

A comparison with previous studies is presented in Table 7. In general, YOLOv7 was more accurate than the version that came before it. Majority of the classes had an improvement in their accuracy with YOLOv7 compared with previous methods. Moreover, the erhu, harp, and saxophone classes had the highest accuracy of over 90% for YOLOv7. The optimal total average accuracy was achieved by YOLOv7, with an accuracy of 86.70%, followed by YOLOv7x with a mAP of 86.10%. Next, Grouplet [10] achieved 85.10% accuracy, and Resnet 50 SPP [41] exhibited an accuracy of 84.66%. Moreover, the clarinet class had the maximum accuracy of 95.70% with Grouplet [10]. The harp class achieved the highest mAP of 98% with Resnet 50 SPP [41]. In terms of size, shape, and appearance, the clarinet and flute are two wind instruments that are very similar to each other. The guitar, violin, and cello are very similar stringed instruments. The sizes of these three musical instruments are different, despite that they are visually similar to one another. The violin is the smallest, the guitar is the second smallest, and the cello is the largest.

**Table 7.** Comparison with previous studies.

| Class Name | Class ID | Grouplet [10] | Resnet 50 SPP [41] | YOLOv7 | YOLOv7x |
|---|---|---|---|---|---|
| Bassoon | 0 | 78.50% | 85.00% | 85.30% | 84.90% |
| Cello | 1 | 87.60% | 81.00% | 86.20% | 87.10% |
| Clarinet | 2 | 95.70% | 89.00% | 82.10% | 77.40% |
| Erhu | 3 | 84.00% | 81.00% | 91.00% | 90.50% |
| Flute | 4 | 87.70% | 82.00% | 83.10% | 81.40% |
| French horn | 5 | 87.70% | 78.00% | 87.10% | 87.30% |
| Guitar | 6 | 93.00% | 79.00% | 87.70% | 85.00% |
| Harp | 7 | 76.30% | 98.00% | 97.30% | 96.80% |
| Recorder | 8 | 84.60% | 85.00% | 83.50% | 80.10% |
| Saxophone | 9 | 82.30% | 93.00% | 91.20% | 90.00% |
| Trumpet | 10 | 87.10% | 85.00% | 81.10% | 85.30% |
| Violin | 11 | 76.50% | 80.00% | 84.40% | 87.10% |
| Average | | 85.10% | 84.64% | 86.70% | 86.10% |

Table 8 shows the results for testing with Dataset 2. Dataset 2 includes images of 30 musical instrument classes for image classification, created by Kaggle (https://www.kaggle.com/datasets/gpiosenka/musical-instruments-image-classification, accessed on 12 April 2023), with 4793 training, 150 testing, and 150 validation images. The images have a size of 224 × 224 × 3 and are in jpg format. Our experiment used seven classes: clarinet, flute, guitar, harp, trumpet, saxophone, and violin. YOLOv7 had the highest average accuracy of 71.9%, with a detection time of 0.014 s. This model was the most accurate and the fastest compared with other models in the experiment. The results of YOLOv7 for recognition in Dataset 2 can be seen in Figure 10.

**Table 8.** Performance with Dataset 2.

| Class Name | YOLOv5n | | YOLOv5s | | YOLOv5m | | YOLOv7 | | YOLOv7x | |
|---|---|---|---|---|---|---|---|---|---|---|
| | Acc (%) | Time (s) | Acc (%) | Time (s) | Acc (%) | Time (s) | Acc (%) | Time (s) | Acc (%) | Time (s) |
| Clarinet | 0.609 | 0.031 | 0.660 | 0.061 | 0.701 | 0.118 | 0.744 | 0.022 | 0.648 | 0.021 |
| Flute | 0.560 | 0.031 | 0.703 | 0.061 | 0.632 | 0.116 | 0.667 | 0.012 | 0.561 | 0.020 |
| Guitar | 0.499 | 0.031 | 0.632 | 0.060 | 0.750 | 0.116 | 0.675 | 0.012 | 0.635 | 0.020 |
| Harp | 0.713 | 0.030 | 0.656 | 0.060 | 0.875 | 0.116 | 0.753 | 0.012 | 0.457 | 0.020 |
| Trumpet | 0.579 | 0.030 | 0.621 | 0.060 | 0.685 | 0.116 | 0.793 | 0.012 | 0.791 | 0.020 |
| Saxophone | 0.668 | 0.030 | 0.644 | 0.060 | 0.625 | 0.118 | 0.638 | 0.012 | 0.817 | 0.020 |
| Violin | 0.689 | 0.031 | 0.798 | 0.060 | 0.693 | 0.116 | 0.763 | 0.012 | 0.652 | 0.020 |
| Average | 0.617 | 0.030 | 0.673 | 0.060 | 0.709 | 0.117 | 0.719 | 0.014 | 0.652 | 0.020 |

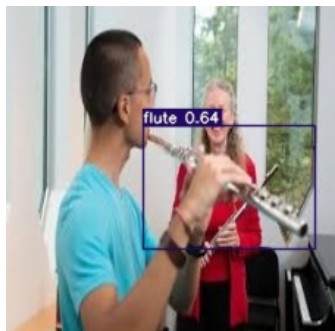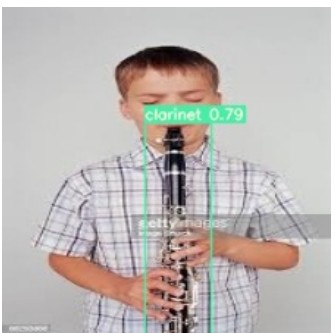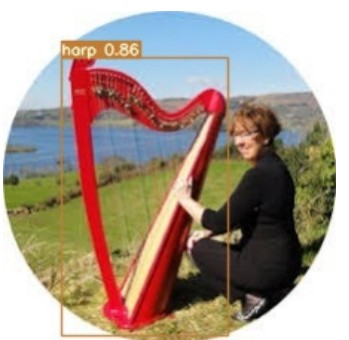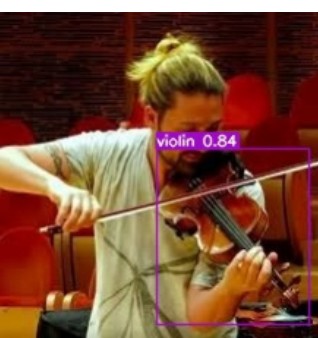

**Figure 10.** Results of YOLOv7 for recognition in Dataset 2.

## 5. Conclusions

In this study, the focus was on distinguishing objects that appear very similar to the human eye. In the process of our investigations, we made use of YOLOv7 and YOLOv5 to determine the identities of various musical instruments. During this investigation, we found several musical instruments that were very similar to one another. YOLOv7 and YOLOv5 are just two of the many backbone architectures and extractor features that our research investigated in conjunction with CNN models for the purpose of object recognition.

According to the findings of our experiment, we were also successful in improving the performance of the system in detecting musical instruments that are similar to one another. YOLOv7 showed a maximum average accuracy of 86.70% compared with previous results: YOLOv7 exhibited a mAP of 86.10%, whereas Grouplet [10] only achieved an accuracy of 85.10%, and Resnet 50 SPP achieved 84.64% [41]. As part of our future research, we hope to find a way to determine whether an image of a musical instrument has the wrong shape. We also plan to use Explainable Artificial Intelligence (XAI) in our future research to help us better understand the images.

**Author Contributions:** Conceptualization, C.D. and A.P.S.C.; data curation, H.J.C.; formal analysis, C.D. and A.P.S.C.; investigation, C.D. and H.J.C.; methodology, C.D.; project administration, C.D.; resources, H.J.C.; software, C.D. and H.J.C.; supervision, A.P.S.C.; validation, C.D., A.P.S.C. and H.J.C.; visualization, H.J.C.; writing—original draft, C.D. and A.P.S.C.; writing—review and editing, C.D. and A.P.S.C. All authors have read and agreed to the published version of the manuscript.

**Funding:** This study was supported by the National Science and Technology Council, Taiwan (Grant number: MOST-111-2637-H-324-001-).

**Institutional Review Board Statement:** Ethical review and approval were waived for this study because we used the public free PPMI dataset. The human faces in the figures are all from the public datasets.

**Informed Consent Statement:** Written informed consent was waived for this study because we used the public free PPMI dataset. The human faces in the figures are all from the public datasets.

**Data Availability Statement:** The PPMI dataset can be found at (http://ai.stanford.edu/~bangpeng/ppmi.html, accessed on 8 January 2023). Dataset 2 was from Kaggle (https://www.kaggle.com/datasets/gpiosenka/musical-instruments-image-classification, accessed on 1 May 2023).

**Acknowledgments:** The authors would like to thank all their colleagues from Chaoyang Technology University, Atma Jaya Catholic University of Indonesia, Satya Wacana Christian University, Indonesia, and all others involved in this research.

**Conflicts of Interest:** The authors declare no conflict of interest.

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
