# Peer review of "Recognizing Similar Musical Instruments with YOLO Models"

_2504-2289, doi:10.3390/bdcc7020094_

Round 1

Reviewer 1 Report

Comments for "Yolov7 for Identifying Similar Musical Instruments based on  Deep Learning"

1. The Yolov7 is a kind of deep learning framework, so the title is redundant.

2. The importance of recognizing musical Instruments is unclear in the introduction

3. Figure 7 only provide AUPRC, the AUROC should be included also.

4. There are two table 4 in the paper

5, Table 3 and first table  4 should be summarized together

6. quality for FIgure 8 should be improved

7. Authors only consider the Yolo V5 and Yolo V7 in their paper rather than other frameworks, they should provide the resaon

Reviewer 2 Report

·       The research issue and objectives can be highlighted in a separate paragraph in the introduction.

·       In section 2.1, the citations such as M. Ju et al., [15] and others are wrong; you don’t cite with initials, so the citations should be as follows: Ju et al. [15] etc. The authors should correct all the in-text citations.

·       I can see the authors only split their dataset into two ratios: testing and training. I believe these datasets are used during the training process. Did the authors test the model? If yes, what dataset did they use? I expect them to use a dataset that has not undergone the training process.

·       I can’t see the model histories; this should be included

·       The dataset size, collection process, and range are not mentioned. The authors should also mention the ratio of the data split as well.

·       The authors should state how the study performance was accessed or evaluated.

·       How did the authors tune the optimal hyperparameter of all models? It should be described clearly.

·       Overall, the English language and presentation style should be improved significantly. There contained a lot of grammatical errors and typos. I suggest you have a colleague proficient in English and familiar with the subject matter review your manuscript or contact a professional editing service.

the English language and presentation style should be improved significantly. There contained a lot of grammatical errors and typos. I suggest you have a colleague proficient in English and familiar with the subject matter review your manuscript or contact a professional editing service.

Round 2

Reviewer 1 Report

authors have fixed all my early concerns

Reviewer 2 Report

All my comments have been attended to